# Linkage of voluntary medical male circumcision clients to adolescent sexual and reproductive health (ASRH) services through Smart-LyncAges project in Zimbabwe: a cohort study

Talent M Makoni ![ORCID] ,[1,2] Pruthu Thekkur,[3,4] Kudakwashe C Takarinda ![ORCID] ,[5] Sinokuthemba Xaba,[5] Getrude Ncube,[5] Nonhlahla Zwangobani,[5] Julia Samuelson,[6] Aveneni Mangombe ![ORCID] ,[5] Simbarashe Mabaya,[2] Talent Tapera,[5] Ronald Matambo,[7] Wole Ameyan,[6] Owen Mugurungi[5]

For numbered affiliations see end of article.

**Correspondence to**
Talent M Makoni;
talent.makoni32@gmail.com

## ABSTRACT

**Objectives** WHO recommended strengthening the linkages between various HIV prevention programmes and adolescent sexual reproductive health (ASRH) services. The Smart-LyncAges project piloted in Bulawayo city and Mt Darwin district of Zimbabwe established a referral system to link the voluntary medical male circumcision (VMMC) clients to ASRH services provided at youth centres. Since its inception in 2016, there has been no assessment of the performance of the referral system. Thus, we aimed to assess the proportion of young (10–24 years) VMMC clients getting 'successfully linked' to ASRH services and factors associated with 'not being linked'.

**Design** This was a cohort study using routinely collected secondary data.

**Setting** All three VMMC clinics of Mt Darwin district and Bulawayo province.

**Primary outcome measures** The proportion of 'successfully linked' was summarised as the percentage with a 95% CI. Adjusted relative risks (aRR) using a generalised linear model was calculated as a measure of association between client characteristics and 'not being linked'.

**Results** Of 1773 young people registered for VMMC services, 1478 (83%) were referred for ASRH services as they had not registered for ASRH previously. Of those referred for ASRH services, the mean (SD) age of study participants was 13.7 (4.3) years and 427 (28.9%) were out of school. Of the referred, 463 (31.3%, 95% CI: 30.0 to 33.8) were 'successfully linked' to ASRH services and the median (IQR) duration for linkage was 6 (0–56) days. On adjusted analysis, receiving referral from Bulawayo circumcision clinic (aRR: 1.5 (95% CI: 1.3 to 1.7)) and undergoing circumcision at outreach sites (aRR: 1.2 (95% CI: 1.1 to 1.3)) were associated with 'not being linked' to ASRH services.

**Conclusion** Linkage to ASRH services from VMMC is feasible as one-third VMMC clients were successfully linked. However, there is need to explore reasons for not accessing ASRH services and take corrective actions to improve the linkages.

### Strengths and limitations of this study

► First study to have assessed the extent of linkages with Smart-LyncAges project.

► No selection bias as we included all the young people (10–24 years) who registered for circumcision services in the voluntary medical male circumcision (VMMC) clinics of two project districts.

► Routine programmatic data of large cohort was used and thus reflected the field realities in project implementation.

► There might have been an underestimation in the proportion of young people 'successfully linked' to adolescent sexual reproductive health (ASRH) services as only those who produced referral slip from VMMC clinic were considered as 'successfully linked'.

► Study was conducted only during 2 months of the year and thus might have failed to account for seasonal variations in seeking ASRH services.

## INTRODUCTION

Young people (aged 10–24 years) are disproportionately affected by the HIV infection, with an estimated 3.9 million living with HIV worldwide in 2017.[1 2] Globally, approximately 1600 youths acquire HIV every day and a youth dies every 10 min because of AIDS-related illness.[1] In contrast to the decline in HIV-related death rates over the years in other age groups, there has been a rise among young people.[3 4] About 84% of young people living with HIV are in sub-Saharan Africa and it is estimated that an additional 7.4 million young people might become infected with HIV in sub-Saharan Africa region alone by 2030.[5]

Unprotected sex is the most common route of HIV infection among young people and is largely due to low knowledge of HIV, early sexual debut, multiple sexual partners and low condom use.[6] This highlights the need for comprehensive sexuality education (CSE) for adolescents before they become sexually active. Realising this, the United Nations General Assembly Special Session on HIV in 2001 recommended access to information, youth-specific HIV education and life skills development to at least 95% of young people.[7] Following this, most sub-Saharan African countries implemented adolescent sexual and reproductive health (ASRH) services. However, recent studies from this region have reported poor utilisation of ASRH services, with only about 21%–51% of adolescents accessing it.[8 9] Acknowledging the importance and poor utilisation of ASRH services, the WHO recommended strengthening the linkages between various HIV prevention and ASRH programmes delivered through general health services.[10]

Zimbabwe, a landlocked sub-Saharan country, has a high burden of HIV, with an estimated prevalence of 14% among people aged 15–49 years.[11] The country has ASRH indicators showing risky behaviours with about 40% girls and 30% of boys having sex before age 18 and 62% of males aged 15–24 not using a condom in their last sexual encounter.[12] HIV prevalence in individuals aged 15–19 years was 3.7% and 4.6% for boys and girls, respectively. Among those 20–24 years, the HIV prevalence was 8% and 10.8% for boys and girls, respectively.[12 13] Thus, since 2009, Zimbabwe has implemented voluntary medical male circumcision (VMMC) and ASRH services as core strategies to limit the HIV burden in young people.

The ASRH programme implemented in 2010 involves the provision of CSE, life skills training, diagnosis and treatment of sexually transmitted infections (STIs) including HIV, family planning counselling, positive masculinity education, vocational training, library services, recreational games and empowerment on rights and responsibility.[14] The ASRH services are delivered at youth-friendly clinics or youth centres. There is anecdotal evidence that the majority of young people do not access ASRH services. Whereas the VMMC programme implemented as part of an HIV prevention strategy has a high utilisation rate.[15] By 2017, about 80% of young boys were covered under VMMC and have had unprecedented contact with the health system. However, the potential of VMMC services as a gateway to additional relevant ASRH services for young people has not been maximised.

The ASRH and VMMC programmes have been implemented as two different vertical national health programmes, not complementing each other in improving service utilisation. In 2016, the WHO supported the Ministry of Health and Child Care to pilot the Smart-LyncAges project to identify approaches to sustain VMMC and improve the complementarity of both services. Through the project, cross-referral mechanisms were established wherein adolescent boys from VMMC are linked to ASRH programmes and vice versa. Though

the Smart-LyncAges project has been piloted as a participatory learning approach since 2016, there is no quantitative information on the extent of successful linkage of clients between VMMC clinics and ASRH services. The information on the extent of successful linkage and factors associated is failure to linkage is necessary to address any deficiencies in the existing referral system prior to scale-up of the project countrywide. In this regard, the study aimed to assess the proportion of young people referred from VMMC clinics successfully linked for ASRH services and factors associated with 'not being linked' in the Smart-LyncAges pilot project sites in Zimbabwe.

## METHODOLOGY
### Study design
This was a cohort study using routinely collected secondary data by the VMMC and ASRH programmes.

### Study setting
#### General setting
Zimbabwe is a landlocked country in sub-Saharan Africa. The country is divided into 10 administrative provinces and has a total of 62 districts. According to the Zimbabwe National Statistics Agency, the country had a projected population of 16.4 million in 2018 and 26% of the total population is adolescents and young adults.[16]

#### Specific setting
The study was conducted in the Bulawayo metropolitan province and Mt Darwin district of Mashonaland Central province, where the VMMC–ASRH linkages project was piloted. Bulawayo is Zimbabwe's second-largest city and has a population of approximately 653000.[16] Mt Darwin is one of the seven districts in the Mashonaland Central province, with a population of about 213000. In the district, the majority of the population resides in rural areas.

#### VMMC services and referral to ASRH
The VMMC programme functions under the National AIDS and TB unit with funding support from external sources, mainly The US Presidents Emergency Plan for AIDS Relief and Bill and Melinda Gates Foundation. The VMMC sites are supported by various non-governmental organisations, which act as implementing partners facilitating service delivery and community mobilisation. The Bulawayo Metropolitan province has two VMMC clinics and the Mount Darwin district has one clinic. The circumcision services are provided at static VMMC clinics and also through outreach clinics in the public health facilities.

Demand generation for VMMC services is done through print and mass media advertisements, school health programmes, roadshows and music galas. At the community level, the community mobilisers, peer-educators or health workers counsel and refer adolescents and young adults for VMMC clinics. The adolescents and young

adults registered for ASRH services are also referred for VMMC. The referral is made to either static VMMC clinics or outreach clinics based on their convenience.

On presentation at a VMMC clinic, the clients are registered with a unique VMMC ID number. The clerk documents the details in the 'client intake form'. During registration, the clerk issues the referral slip to all the clients who are not previously registered for ASRH services. The clients are advised to register themselves for ASRH services in their preferred youth centre and referral slip is provided. The referral slip is created in triplicate, one slip given to the adolescent, one maintained in the referral file at the VMMC clinic and the other slip sent to the peer-educator of the service area from where the adolescent has come.

A nurse counsellor offers group education and individual counselling to clients registered for VMMC. During counselling, the importance of registering for ASRH services is emphasised. The nurse counsellors highlight the various ASRH services present in the youth centres and also provide detailing on the benefits of enrolling for ASRH services at youth centres. The nurse counsellors provide HIV testing and a preoperation examination to assess the eligibility for circumcision surgery. Those adolescents with diabetes, keloids and haemophilia are considered not eligible for surgery. Among those eligible for surgery, either dorsal slit or forceps guided procedure is performed by either a trained and qualified nurse circumciser or doctor surgeon. Clients are advised to make follow-up visits on day-2 and day-7 after the procedure. All the details on surgery, follow-up and adverse events are recorded in the 'client intake form'.

### ASRH services and registration

The ASRH services are provided at the youth centres. The Bulawayo metropolitan province has 17 youth centres and the Mount Darwin district has two youth centres. Each youth centre is staffed with a youth health advisor (a registered general nurse), a youth facilitator/recreational officer (a social worker) and five peer-educators.

On arrival at the youth centres, the VMMC clients submit their VMMC–ASRH referral slips. The youth facilitator/recreational officer registers the adolescents for ASRH services and documents the VMMC ID number in the VMMC–ASRH linkages register. The services received by the adolescent during their initial visit are documented in the VMMC–ASRH linkage register. The referral slip received from the adolescent is stored in the referral box maintained at the youth centre.

### Study population

All adolescents and young adults (10–24 years) registered in VMMC clinics of Mt Darwin district and Bulawayo city during October and November 2018 were included in the study. Those who were previously registered for ASRH services prior to accessing VMMC services were excluded. The sample size was not calculated and there was no sampling as all the adolescents and young adults in both

the pilot districts of the Smart-LyncAges project during study reference period were included.

### Data variables, sources of data and data collection

We extracted details including VMMC identification number, age, education status, mode of referral to VMMC clinic, type of VMMC clinic, HIV status, date of referral to ASRH services, date of registration at VMMC, reason for circumcision, eligibility for circumcision, circumcision status, method of circumcision, status of day-2 post-op visit, status of day-7 post-op visit and adverse events following circumcision within 42 days from the VMMC 'client intake form' maintained at VMMC clinic. The details on registration and date of registration for ASRH services were extracted from the VMMC–ARSH linkage register maintained at youth centres. The information on registration for ASRH services extracted from the VMMC–ASRH register was validated using the referral slips maintained in the referral box of youth centres. Those individuals linked for ASRH services within 3 months of receiving referral slip were considered as 'successfully linked' to ASRH services.

The data were extracted in March 2019 using two separate structured data extraction proformas designed to extract data from VMMC 'client intake form' and VMMC–ARSH linkage register. Thus, for each of the study, participants registered during October and November 2018, the linkage status was ascertained only after a minimum of 90 days of follow-up. The principal investigator field-tested data extraction proformas and modified them before data extraction.

### Data entry and analysis

Data were double entered and validated using EpiData Entry software (EpiData Association, Odense, Denmark). Two separate data entry structures were used to enter data from two proformas. The two data structures were then merged using the unique 'VMMC ID number'. The final merged data file was used for analysis.

Data were analysed using Stata V.12.0. Sociodemographic, HIV status and clinical characteristics were summarised using percentages. The age of the participants was summarised with mean and SD. The proportion of participant 'successfully linked' to ASRH was summarised as percentages with 95% CI. The duration between referral to 'successfully linked' to ASRH was summarised with median and IQR.

The association between sociodemographic and baseline clinical characteristics with 'not being linked' for ASRH service was assessed using bivariate log-binomial regression. A generalised linear model (Poisson regression) with robust variance estimates was used for multivariate regression as the log-binomial model did not converge.[17 18] Initially, all the variables with p value<0.25 in the bivariate model was included in the multivariate model. Later, the variables with a variance inflation factor of more than 10 were removed from the final model and

adjusted relative risks (aRR) with 95% CI were expressed as measure of association.

## Patient and public involvement

Principal investigator (PI) and the data collectors did not interact directly with the young people availing services from VMMC clinics during this retrospective record review. The PI worked with the healthcare staff and peer-educators of both the VMMC and ASRH clinics included in the study. Findings from this study will help the Smart-LyncAges project to assess the performance in linkages and also gives insight on deficiencies to be fixed prior to country wide scale-up.

## RESULTS

In total, 1773 young people (10–24 years) were registered for VMMC services during the study reference period. Of the total, 1478 (83%) had not been previously registered in the ASRH clinic and were included in the study. The mean (SD) age of study participants was 13.7 (4.3) years and 427 (28.9%) were out of school. Of the 1478 study participants, 1032 (69.8%) were referred for VMMC by community mobiliser and 1230 (83.2%) reported HIV prevention as the reason for seeking circumcision service. Of the total, 6 (0.3%) had HIV infection. The characteristics of study participants are depicted in table 1.

Of the 1478 study participants, 1461 (99%) were eligible for circumcision surgery, of which, 1443 (99%) underwent circumcision (figure 1). Of those who underwent circumcision, 477 (33.1%) had the surgery performed at an outreach site. Of the 1443 participants, 1379 (95.6%) and 1025 (71%) attended day-2 and day-7 review visits, respectively. In total, 20 (1.4%) developed mild adverse events following circumcision and 3 (0.2%) each developed moderate and severe adverse events (table 2).

Of the 1478 study participants referred for ASRH services, 463 (31.3%, 95% CI: 29.0% to 33.8%) successfully linked for ASRH services at 'youth centre' within 3 months of referral (figure 1). Among those who were successfully linked, the median (IQR) duration from referral to getting linked at 'youth centre' was 6 (0–56) days. On adjusted analysis, referral from Bulawayo circumcision clinic (aRR: 1.5 (95% CI: 1.3 to 1.7)) and undergoing circumcision at outreach sites (aRR: 1.2 (95% CI:1.1 to 1.3)) were associated with not being linked for ASRH services after referral from circumcision clinic. The AIC and BIC of the model were 2799.2 and 2926.3, respectively. The LR test was significant compared with constant model (p=0.014). The association between participant's characteristics and not being linked is shown in table 3.

## DISCUSSION

This is the first study in assessing the extent of successful referral among adolescents and young men between VMMC clinics and ASRH services under the

**Table 1** The demographic characteristics, reason for circumcision and HIV status of adolescents and young adults registered for VMMC services during October and November, 2018 in selected health facilities of Zimbabwe, n=1478

| Characteristics | Categories | Frequency | (%) |
|---|---|---|---|
| Total | | 1478 | (100) |
| Age (in years) | 10–14 | 980 | (66.3) |
| | 15–19 | 314 | (21.2) |
| | 20–24 | 184 | (12.3) |
| Education status | Out of school | 427 | (28.9) |
| | Primary education | 899 | (60.8) |
| | Secondary education | 152 | (10.3) |
| Health facility | Mt Darwin | 542 | (36.7) |
| | Lobengula MC clinic | 587 | (39.7) |
| | Bulawayo MC clinic | 349 | (23.6) |
| Referred by* | Friend or partner | 37 | (2.5) |
| | Health worker | 216 | (14.6) |
| | Community mobiliser | 1032 | (69.8) |
| | Others | 139 | (9.4) |
| | Missing | 54 | (3.7) |
| Reasons for circumcision† | HIV prevention | 1230 | (83.2) |
| | Sexual pleasure | 248 | (16.8) |
| | STI prevention | 736 | (49.8) |
| | Hygiene | 1196 | (80.9) |
| | Sociocultural reasons | 51 | (3.5) |
| HIV status | Positive | 6 | (0.3) |
| | Negative | 1472 | (99.7) |

*The person who referred the client to VMMC clinic.
†Multiple responses are possible.
MC, male circumcision; STI, sexually transmitted infection; VMMC, voluntary medical male circumcision.

Smart-LyncAges project implemented in Zimbabwe. About 31% of young males (10–24 years) referred from VMMC clinics successfully linked to ASRH services at youth centres within 3 months of referral. Young people referred from the outreach sites and Bulawayo male circumcision clinic had significantly higher rates of not being linked for ASRH services.

Globally, there is limited literature on linkages between various HIV preventive strategies like VMMC services and ASRH services. A study from Zambia reported that, with enhanced counselling and referral from the community, uptake of ASRH services like HIV testing and counselling,

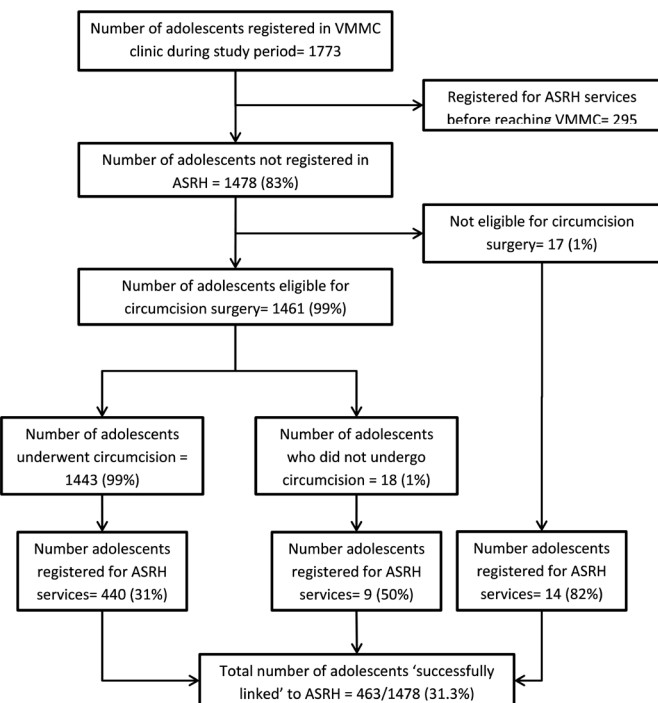

**Figure 1** Flow-chart depicting the adolescents eligible for referral to ASRH services, circumcision status and linkage to ASRH services among those registered at selected VMMC clinics of Zimbabwe during October to November 2018. ASRH, adolescent sexual reproductive health; VMMC, voluntary medical male circumcision.

**Table 2** The surgical and follow-up details of adolescents and young adults who underwent circumcision during October and November, 2018 in selected male circumcision clinics of Zimbabwe, n=1443

| Characteristics | Categories | Frequency | (%) |
|---|---|---|---|
| Total | | 1443 | (100.0) |
| Method of circumcision | Dorsal slit | 1439 | (99.7) |
| | Forceps guided | 4 | (0.3) |
| Service delivery point* | Static site | 966 | (66.9) |
| | Outreach site | 477 | (33.1) |
| Circumciser† | Nurse practitioner | 1418 | (98.3) |
| | Doctor | 25 | (1.7) |
| Attended post-op day-2 | Yes | 1379 | (95.6) |
| | No | 64 | (4.4) |
| Attended post-op day-7 | Yes | 1025 | (71.0) |
| | No | 418 | (29.0) |
| Adverse event | None | 1417 | (98.2) |
| | Mild | 20 | (1.4) |
| | Moderate | 3 | (0.2) |
| | Severe | 3 | (0.2) |

*The circumcision in VMMC clinics is considered as static sites. The circumcision done at other public health facilities in a camp mode by staff of VMMC clinics is considered as outreach sites.
†Circumciser is the one who conducted the circumcision.
VMMC, voluntary medical male circumcision.

family planning counselling and VMMC services was less than 20% within 6 months.[19] Also, previous reviews have reported that demand generation activities in the community, improving accessibility and quality of ASRH services, improve ASRH-related knowledge but may not increase the utilisation of specific ASRH services.[20–22] The potential reason for underutilisation is that the preventive strategy like ASRH may not be considered as an immediate need by young people.[22–24] Considering this, the successful linkage (uptake) of about one in three of the VMMC clients to ASRH services through the SmartLyncAges project (as they produced referral slips) is promising.

The clients referred from the Bulawayo MC clinic had a slightly higher risk of not being linked to ASRH services. A potential reason for this finding is that the MC clinic is situated in the eastern suburbs of the city where people of high socioeconomic status reside. All the 15 youth centres were located in western suburbs, which are largely populated with people from low socioeconomic status. It is possible that the recreational activities available under ASRH services do not appeal to the technologically savvy urban youths from the elite suburbs. Furthermore, distance to youth centres might have limited the clients from the eastern suburbs to go and register for ASRH services. Similarly, clients accessing VMMC services at outreach clinics were not being linked to ASRH services. The focus of outreach clinics is to conduct surgery in hard to reach areas. Though the referral slips were issued,

there would not be space and time to counsel on ASRH services comprehensively. Also, the youth centres in Mt Darwin were mainly located in district centres and might not be accessible to clients of outreach clinics conducted in hard to reach areas. Thus, though not assessed objectively, there might be a rural–urban divide in linkages. Although the actual numbers were small, a high proportion of clients ineligible for circumcision or experiencing an adverse event were referred and successfully linked, suggesting that they have additional service needs.

The study had several strengths. First, the study was conducted within the routine programmatic setting, reflecting the field realities and using routine data. Second, all the young males attending the VMMC clinics during the study reference period were included and thus, there was no sampling bias. Third, the study had a large sample size and thus, the estimate of 'successfully linked' was precise. Fourth, the continuous supportive supervision was rendered during the study period to ensure completeness in recording under the project setting and thus, the 'missing data' was limited. Fifth, the registration for ASRH services was validated by cross-checking

**Table 3** Demographic, HIV status and surgical characteristics associated with not getting linked for ASRH services within 3 months of referral from the selected VMMC health facilities in Zimbabwe, October and November 2018

| Characteristic | Category | Total | Not linked to ASRH, n (%)* | Linked to ASRH, n (%) | Unadjusted RR (95% CI) | Adjusted RR (95% CI)† | P value |
|---|---|---|---|---|---|---|---|
| Total | | 1478 | 1015 (68.7) | 463 (31.3) | | | |
| Age (in years) | 10–14 | 980 | 671 (68.5) | 309 (31.5) | 1.1 (1.0 to 1.2) | 1.0 (0.9 to 1.2) | 0.828 |
| | 15–19 | 314 | 200 (63.7) | 114 (36.3) | 1 | 1 | |
| | 20–24 | 184 | 144 (78.3) | 40 (21.7) | 1.2 (1.1 to 1.4) | 1.1 (1.0 to 1.3) | 0.168 |
| Education | Out of school | 427 | 297 (69.6) | 130 (30.4) | 1.1 (1.0 to 1.3) | 1.0 (0.9 to 1.2) | 0.560 |
| | Primary | 899 | 622 (69.2) | 277 (30.8) | 1.1 (1.0 to 1.2) | 1.0 (0.9 to 1.2) | 0.632 |
| | Secondary | 152 | 96 (63.2) | 56 (36.8) | 1 | 1 | |
| Referred by | Friend/partner | 37 | 22 (59.5) | 15 (40.5) | 1 | 1 | |
| | Health worker | 216 | 164 (75.9) | 52 (24.1) | 1.3 (1.0 to 1.7) | 1.2 (0.9 to 1.6) | 0.207 |
| | Community mobiliser | 1032 | 694 (67.3) | 338 (32.7) | 1.1 (0.9 to 1.5) | 1.1 (0.8 to 1.5) | 0.453 |
| | Others | 139 | 106 (76.3) | 33 (23.7) | 1.3 (1.0 to 1.7) | 1.0 (0.8 to 1.4) | 0.849 |
| | Missing | 54 | 29 (53.7) | 25 (46.3) | 0.9 (0.6 to 1.3) | 1.2 (0.8 to 1.6) | 0.345 |
| Health facility | Mt Darwin | 542 | 330 (60.9) | 212 (39.1) | 1 | 1 | |
| | Lobengula MC | 587 | 386 (65.8) | 201 (34.2) | 1.1 (1.0 to 1.2) | 1.1 (1.0 to 1.3) | 0.087 |
| | Bulawayo MC | 349 | 299 (85.7) | 50 (14.3) | 1.4 (1.3 to 1.5) | 1.5 (1.3 to 1.7) | <0.001 |
| HIV status | Positive | 6 | 5 (83.3) | 1 (16.7) | 1.2 (0.8 to 1.7) | 1.3 (0.9 to 1.9) | 0.117 |
| | Negative | 1472 | 1010 (68.6) | 462 (31.4) | 1 | 1 | |
| Reasons for MC | | | | | | | |
| HIV prevention | Yes | 1230 | 865 (85.2) | 365 (14.8) | 1.2 (1.0 to 1.3) | 1.0 (0.9 to 1.1) | 0.800 |
| | No | 248 | 150 (60.5) | 98 (39.5) | 1 | 1 | |
| Sexual pleasure | Yes | 248 | 158 (63.7) | 90 (36.3) | 0.9 (0.8 to 1.0) | 0.9 (0.8 to 1.1) | 0.593 |
| | No | 1230 | 857 (69.7) | 373 (30.3) | 1 | 1 | |
| STI prevention | Yes | 736 | 523 (71.1) | 213 (28.9) | 1.1 (1.0 to 1.1) | 1.0 (0.9 to 1.1) | 0.980 |
| | No | 742 | 492 (66.3) | 250 (33.7) | 1 | 1 | |
| Hygiene | Yes | 1196 | 847 (70.8) | 349 (29.2) | 1.2 (1.1 to 1.3) | 1.1 (1.0 to 1.2) | 0.183 |
| | No | 282 | 168 (59.6) | 114 (40.4) | 1 | 1 | |
| Sociocultural | Yes | 51 | 32 (62.8) | 19 (37.2) | 0.9 (0.7 to 1.1) | 0.9 (0.7 to 1.1) | 0.407 |
| | No | 1427 | 983 (68.9) | 444 (31.1) | 1 | 1 | |
| Circumcision§ | Not eligible | 17 | 3 (17.7) | 14 (82.3) | 1 | 1 | |
| | Circumcised | 1443 | 1003 (69.5) | 440 (30.5) | 3.9 (1.4 to 11) | 2.9 (0.6 to 8.8) | 0.062 |
| | Not circumcised | 18 | 9 (50.0) | 9 (50.0) | 2.8 (0.9 to 8.7) | 2.3 (0.8 to 7.1) | 0.142 |
| Circumciser§ | Nurse | 1418 | 989 (69.8) | 429 (30.2) | 1.2 (0.9 to 1.8) | 1.2 (0.8 to 1.8) | 0.283 |
| | Doctors | 25 | 14 (56.0) | 11 (44.0) | 1 | 1 | |
| Service delivery point§ | Static | 966 | 66 (65.8) | 900 (34.2) | 1 | 1 | |
| | Outreach | 477 | 367 (76.9) | 110 (23.1) | 1.2 (1.1 to 1.2) | 1.2 (1.1 to 1.3) | <0.001 |
| Day-2 follow-up§¶ | Yes | 1379 | 965 (70.0) | 414 (30.0) | 1.2 (1.0 to 1.4) | | |
| | No | 64 | 38 (59.4) | 26 (40.6) | 1 | | |
| Day-7 follow-up§ | Yes | 1025 | 732 (71.4) | 293 (28.6) | 1.1 (1.0 to 1.2) | 0.9 (0.8 to 1.0) | 0.070 |
| | No | 418 | 271 (64.8) | 147 (35.2) | 1 | 1 | |
| Adverse event‡§¶ | Yes | 26 | 11 (42.3) | 15 (57.7) | 1 | | |
| | No | 1417 | 992 (70.0) | 425 (30.0) | 1.7 (1.1 to 2.6) | | |

*Row percentage.
†Using generalised linear model (Poisson regression).
‡Mild, moderate and severe were clubbed as 'Yes'.
§Applicable only for those who underwent circumcision.
¶These factors were removed from the final model as they had variance inflation factor of more than 10.
ASRH, adolescent sexual reproductive health; MC, male circumcision; RR, relative risk; STI, sexually transmitted infection; VMMC, voluntary medical male circumcision.

the stored referral slips along with information extracted from the VMMC–ASRH linkage register, and thus it enhanced the validity of the outcome ascertainment. Sixth, the STrengthening the Reporting of OBservational Studies in Epidemiology (STROBE) guidelines was used to report the study findings.[25]

The study had a few limitations. First, the registration for ASRH services was ascertained based on the VMMC–ASRH linkage register. Only those who produced VMMC referral slips during registration were listed in the VMMC–ASRH register. Thus, those participants who might have missed their referral slips before registration for ASRH services could have been misclassified as 'not linked'. This could have led to the underestimation of the percentage of 'successfully' linked. Second, the study was conducted only during 2 months of the year and thus failed to account for seasonal variations in seeking services. During the festive month of December, the youth centres were closed due to holidays. Also, during December, the youth centres in the Mt Darwin region were closed for 2 weeks due to the cholera outbreak in the region. There might have been an underestimation of 'successfully' linked due to the non-functioning of youth centres during the study follow-up period. However, as there are no estimates of the extent of underestimation, we failed to account for it in estimating the percentage of successfully linked. Third, the pathways for seeking ASRH services and the contribution of peer-educators in reaching the ASRH services were not assessed. Thus, the study failed to document the pathways with the highest successful linkage rate, which could have helped in replicating the model elsewhere. Fourth, the potential confounders like distance from youth centres, socioeconomic status, parents' willingness and adolescents' willingness were not captured and included in the adjusted analysis. The inclusion of these variables could have improved the validity of the model. Fifth, the study represents select group of young people who came for VMMC clinics in the project pilot districts. Thus, the generalisability of the study results is limited. Finally, the study did not explore the reasons for not being linked to ASRH services among those referred.

The study has a few programmatic implications and recommendations. First, the percentage of 'successfully linked' between VMMC clinics and ASRH services was promising. The Smart-LyncAges project can be scaled up to improve the utilisation of ASRH services. However, there is a need for qualitative research to explore the facilitators and barriers to getting linked to ASRH services. This information can help to improve the linkages and effectiveness of the programme.

Second, with the existing records maintained under the project, cohort monitoring of the performance of the referral system was feasible. The 'proportion of those referred from the facility successfully linked for ASRH services' can be introduced as an indicator in the monthly report of each VMMC clinic. This would enable the programme managers to monitor the performance of the referral system better than the absolute number of referred and linked, which is currently being monitored.

Third, the young adolescents seeking VMMC services from outreach clinics at peripheral public health facilities were not getting linked for ASRH services, implying the distance to youth centres was a potential barrier. Solutions should be person-centred, such as colocation of ASRH and VMMC services, improved accessibility with services decentralised to peripheral public health facilities and enhanced training of existing general health staff to deliver youth-friendly services.

Fourth, about 70% of clients came back to avail follow-up services on day-7 postcircumcision. This opportunity can be used to educate further and to reinforce the importance of ASRH services. The clients might be more responsive to the information since the circumcision procedure is already completed.

## CONCLUSION

The rate of young males referred from VMMC clinics and successfully linked to ASRH services was promising. Those referred from the outreach sites and clinics in Bulawayo had significantly higher rates of 'not being linked'. However, there is a need to explore the reasons for clients not accessing the ASRH services and take corrective actions to improve the linkages.

**Author affiliations**
[1]AIDS and TB Unit, Ministry of Health and Child Care, Harare, Harare, Zimbabwe
[2]World Health Organization Regional Office for Africa, Harare, Harare, Zimbabwe
[3]Centre for Operational Research, International Union Against Tuberculosis and Lung Disease, Paris, France
[4]Centre for Operational Research, The Union South-East Asia Office, New Delhi, India
[5]Ministry of Health and Child Care, Harare, Harare, Zimbabwe
[6]World Health Organization, Geneve, GE, Switzerland
[7]The Union Zimbabwe, Harare, Harare, Zimbabwe

**Acknowledgements** This research was conducted through the Structured Operational Research and Training Initiative (SORT IT), a global partnership led by the Special Programme for Research and Training in Tropical Diseases at the World Health Organisation (WHO/TDR). The training model is based on a course developed jointly by the International Union Against Tuberculosis and Lung Disease (The Union) and Medécins sans Frontières (MSF). The specific SORT IT programme which resulted in this publication was implemented by the Centre for Operational Research, The Union, Paris, France. Mentorship and the coordination/facilitation of this particular SORT IT workshop was provided through the Centre for Operational Research, The Union, Paris, France; the Department of Tuberculosis and HIV, The Union, Paris, France; The Union, Zimbabwe Office; The Union, South East Asia Office; University of Washington, School of Public Health, Department of Global Health, Seattle, Washington, USA; National Institute for Medical Research, Muhimbili Centre, Dar es Salaam, Tanzania; and AIDS & TB Department, Ministry of Health & Child Care, Harare, Zimbabwe.

**Contributors** TMM was the principal investigator; PT and KCT were the SORT IT course mentors; OM, WA, JS, SX and GN were the senior authors. TMM, NZ, AM, SM, TT and RM were involved in data collection; TMM, PT and KCT analysed the data and prepared the first draft of the paper. All authors were involved in conception, design, inference of results, providing critical review and approval of the final manuscript.

**Funding** The training course under which this study was conducted was funded by: the United Kingdom's Department for International Development (DFID); The

Global Fund to Fight AIDS, Tuberculosis and Malaria (GFATM) and the WHO. The WHO also provided funding support for conduct of the study.

**Competing interests**  None declared.

**Patient and public involvement**  Patients and/or the public were not involved in the design, or conduct, or reporting or dissemination plans of this research.

**Patient consent for publication**  Not required.

**Ethics approval**  The ethics approval was obtained from the Medical Research Council of Zimbabwe (MRCZ/E229) and Ethics Advisory Group of the International Union Against Tuberculosis and Lung Disease, Paris, France (58/18). Permission was sought from the Ministry of Health and Child Care program officials for extracting the routine data collected at VMMC and ASRH centres.

**Provenance and peer review**  Not commissioned; externally peer reviewed.

**Data availability statement**  Data are available in a public, open access repository. Technical appendix, statistical code, and dataset available from the https://www.dropbox.com/sh/cm03olkw3qj8j4x/AAArNAzjweJ1iw0UHq86JlFwa?dl=0.

**ORCID iDs**
Talent M Makoni http://orcid.org/0000-0003-4303-5521
Kudakwashe C Takarinda http://orcid.org/0000-0002-2980-7735
Aveneni Mangombe http://orcid.org/0000-0002-4057-3378

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
