## [Reviewer comments · BMJ Open]

ARTICLE DETAILS

TITLE (PROVISIONAL)	Linkage of Voluntary Medical Male Circumcision Clients to Adolescent Sexual and Reproductive Health (ASRH) services through Smart-LyncAges project in Zimbabwe: A cohort study
AUTHORS	Makoni, Talent; Thekkur, Pruthu; Takarinda, Kudakwashe; Xaba, Sinokuthemba; Ncube, Getrude; Zwangobani, Nonhlahla; Samuelson, Julia; Mangombe, Aveneni; Mabaya, Simbarashe; Tapera, Talent; Matambo, Ronald; Ameyan, Wole; Mugurungi, Owen

VERSION 1 – REVIEW

REVIEWER	Kennedy Otworld Perinatal HIV Research Unit, Chris Hani Academic Hospital, University of the Witwatersrand, Johannesburg, South Africa
REVIEW RETURNED	06-Aug-2019

GENERAL COMMENTS	Abstract: The abstract is well summarized. The authors need to clarify what was done and the findings, i.e. one of the objectives was to check for “factors associated with not being linked”, however, in the results part of the abstract “factors associated with not being registered” was reported. Describe the population characteristics in one-line e.g what was their age. Introduction: The authors presented what is happening globally, in Africa then narrowed to Zimbabwe and why the study was conducted. The literature used is relevant and includes recent works. However, the introduction is too long given that only three tables of results have been presented Objectives: The main objectives of the study is stated as assessing the proportion of VMMC clients getting ‘successfully linked’ to ASRH services and factors associated with not being linked. However, the authors keep switching between not linked and not registered for ASRH whereas the data used for final analysis included only those who were not registered for ASRH prior to the study period.
---

	Methodology: Study design and participants/population  • How was the sample size selected? • What sampling method was used? • Eligibility of the participants into study need to be included, for instance, how many were recruited (overall)? How many excluded and why? How many were considered for analysis? Did they all consent? (Line 213 to 216 can be moved under this section) Measures  • Clearly define all variables/measures described in the table of results • How were they measured? Statistical analysis:  • In the statistical methods section, the authors have not described all the statistical methods reported in the tables i.e. mean, median, etc. • Which variables were considered at the univariate and multivariate levels? • Clarify why you used RR? • The multivariate model has a lot of variables. This may lead to collinearity in the model fitted. How was model fit assessed? It appears the authors only reported the full model but a model selection procedure should have been applied to determine the final reduced model. Results:  • Overall frequencies and percentages presented in table 1 and those not linked in table 3 for all variables considered, I think it will be much easy to read if one table consists of overall, those linked and not linked • For the proportion of those “successfully linked”: I do not see the need for reporting the confidence interval Discussion: The authors present only one page of results and four pages of discussion. It would be helpful to keep the discussion short, clear
--	--

	and linked to the results presented without repetitions. Overall: The authors need to clarify in all stages in the study whether they trying to model factors associated with not linked or not registered for ASRH Consider deleting lines 231 to 234 Line 270 to 272: The disposition flow chart is unclear without arrows showing directionality. I am not sure if this is because of the BMJ Open system or the way the authors prepared and presented it. Line 272 to 273 “the mean (SD) age of study participants was 13.7 (4.3) years”: results presented here are not shown in any of the tables Line 285 to 286 “the median (IQR) duration from referral to getting registered at ‘youth centre’ was 6 (0-56) days”: results presented here are not shown in any of the tables Line 298 to 299 “Young people referred from the outreach sites and Bulawayo male circumcision clinic had significantly higher rates of not being registered for ASRH services.”: Does this refer to the overall or only the sample size considered in the analysis? Line 315 Reword this sentence to read better “. . . youth centres are however were located . . .” The authors need to be consistent with the referencing style in using () and [].
--	--

REVIEWER	Makumbi Fred Makerere University, Uganda
REVIEW RETURNED	10-Aug-2019

GENERAL COMMENTS	i) Data analysis plan seems inadequate ii) Justify why use log-binomial, and was this model able to converge? iii) I suggest you use one approach for both bivariate and multivariable regression, either log-binomial or glm with poisson iv) It seems your work would have benefited from some interaction, but these were never tested, say between education and age, or rural/urban residence and education check literature review and think through models of health seeking behaviors iv) It is not clear if you considered all clients for this analysis or just those with at least 3 months since VVMC, because your outcome could only be determined after 3 months v) I think you should maintain one clear outcome and also determine factors associated with that outcome. In this write up the outcome does not have associated factors vi) For the discussion section, you should be able to also fully discuss if these results were in an unexpected direction and why this was so. I would personally expect to see higher levels of linkage, because these are already seeking a health service by going for VVMC! However, I also note that most of your clients were young
--

	(10-14). Discuss if the VVMC staff were able to bring up issues of ASRH to enable these young clients seek such services? I also think you need to clearly discuss the issues of access to the ASRH access in terms of distance but also how health workers at such sites handle young clients
--	---

VERSION 1 – AUTHOR RESPONSE

Reviewer: 1

Reviewer Name: Kennedy Otwombe

Institution and Country: Perinatal HIV Research Unit, Chris Hani Academic Hospital, University of the Witwatersrand, Johannesburg, South Africa Please state any competing interests or state 'None declared': None declared

Note: *All the line numbers mentioned in the response relates to that in the main document with track changes*

Abstract

Comment-1: The abstract is well summarized. The authors need to clarify what was done and the findings, i.e. one of the objectives was to check for “factors associated with not being linked”, however, in the results part of the abstract “factors associated with not being registered” was reported.

Response: Thank you for appreciating the abstract. Thank you for highlighting inconsistency in labelling our primary outcome. Our analysis included only those who were not registered for ASRH services and thus, the ‘not being linked’ is the appropriate word here. We acknowledge that this inconsistency created confusion with the interpretation of results.

To overcome this, throughout the manuscript we have consistently used ‘successfully linked’ and ‘not being linked’ as our outcomes of interest. Changes made in line number 71.

Introduction

Comment-2: The authors presented what is happening globally, in Africa then narrowed to Zimbabwe and why the study was conducted. The literature used is relevant and includes recent works. However, the introduction is too long given that only three tables of results have been presented.

Response: Thank you for the suggestion. We acknowledge that the introduction was too long with 774 words. Now, we have reduced the content without harming the flow of the information which we had presented in our previous version. We have cut about 100 words in the introduction and now the word count of introduction is 670. Changes made in line number 92 to 155.

Objectives

Comment-3: The main objectives of the study is stated as assessing the proportion of VMMC clients getting ‘successfully linked’ to ASRH services and factors associated with not being linked. However,

the authors keep switching between not linked and not registered for ASRH whereas the data used for final analysis included only those who were not registered for ASRH prior to the study period.

Response: Thank you highlighting this inconsistency. As mentioned earlier, our analysis included only those who were not registered for ASRH services. Thus, the 'not being linked' is the appropriate word here. We acknowledge that this inconsistency created confusion with the interpretation of results. To overcome this, throughout the manuscript we have consistently used 'successfully linked' and 'not being linked' as our outcomes of interest.

Methods

Comment-4: How was the sample size selected? What sampling method was used? Eligibility of the participants into study need to be included, for instance, how many were recruited (overall)? How many excluded and why? How many were considered for analysis? Did they all consent? (Line 213 to 216 can be moved under this section)

Response: We included all the adolescents and young adults who registered for VMMC services during October and November, 2018 in both the Smart-LyncAges project pilot districts. Thus, we did not adopt any sampling technique as all Smart-LyncAges districts and all the adolescents/young adults availing VMMC services in these districts during reference period were included. The sample size was not calculated as it was an operational research study built on retrospective programme data.

However, we acknowledge that this was not explicitly reported in the manuscript. Now we report the same as 'Sample size was not calculated and there was no sampling as all the adolescents and young adults in both the pilot districts of Smart-LyncAges project during study reference period were included' in line number 227 to 229 .

The ethics committees provided waiver from participant consent as the details were extracted from the routinely maintained forms under Smart-LyncAges project. We have explained the participants, exclusion and reasons for exclusion under the results section as per STROBE checklist (Line number 292 to 303). We wish to retain it under the results section of the manuscript.

Comment-5: Clearly define all variables/measures described in the table of results How were they measured?•

Response: As all the variables were extracted from the existing records. The details of source of data for each of the variable included in the study is explained in line number 233 to 243.

As suggested, wherever the description of variable is not clear, we have described the variables as footnote in Table-1 and Table-2 .

Comment-6: In the statistical methods section, the authors have not described all the statistical methods reported in the tables i.e. mean, median, etc.

Response: Thank you for noting this deficiency in reporting. Now we have added two lines, 'The age of the participants was summarized with mean and standard deviation (SD)' and 'The duration between referral to 'successfully linked' to ASRH was summarized with median and inter-quartile range (IQR)'. The changes have been made in line number 258 to 261.

Comment-7: Which variables were considered at the univariate and multivariate levels? Clarify why you used RR? The multivariate model has a lot of variables. This may lead to collinearity in the model fitted. How was model fit assessed? It appears the authors only reported the full model but a model selection procedure should have been applied to determine the final reduced model.●

Response: We included all the variables as this was an explanatory model and the interest here was to assess the factors which are associated with the outcome in contrary to reduced predictive models. Thus, we did not adopt any model selection procedure.

The relative risks (RR) was used as this was a cohort study. Also, the relative risks are better measure of association than the odds ratios which overestimate the risk. The odds ratios provide a good estimate of risk in case of rare outcomes (<5%), they overestimate the magnitude of association, whenever the outcomes are not rare, as it is the case in our study (not linked was ~69%).

Thank you for highlighting on the potential collinearity between the variables. We have checked for the collinearity. Now the model has only variables with variance inflation factor of less than 10 and factors like circumcision status which are considered in spite of collinearity. However, this did not change the association between the significant factors which were reported in the previous version. We really appreciate the reviewer for this comment which helped us to improve the power of the model by reducing collinearity. We have made changes in the analysis plan based on your inputs in the line number 262 to 270. Changes have also been made in the estimates reported in the Table-3.

Results

Comment-8: Overall frequencies and percentages presented in table 1 and those not linked in table 3 for all variables considered, I think it will be much easier to read if one table consists of overall, those linked and not linked

Response: Thank you for this suggestion. We have made a prompt attempt to combine two tables together. However, we feel the tables become too crowded and confusing with column percentage, row percentage and relative risks. Thus, we retain the tables as it was in previous version.

Comment-9: For the proportion of those “successfully linked”: I do not see the need for reporting the confidence interval

Response: We differ from the opinion of the reviewer here and wish to retain the confidence interval. The confidence interval is given as a measure of precision and error estimate for generalization. Though we have included all the participants (universal sample), the participants are sampled by the study reference period.

Discussion

Comment-10: The authors present only one page of results and four pages of discussion. It would be helpful to keep the discussion short, clear and linked to the results presented without repetitions.

Response: Thank you for the suggestion. We have structured the discussion under the following headings- Key findings of the study (1 paragraph), Comparison and reasons for the key findings (2 paragraphs), strengths (1 paragraph), limitations (1 paragraph) and implications/recommendation (4 paragraphs).

We agree that we might have stretched a bit in the implications and recommendation section than the usual academic research. This was intentional as the findings of this first assessment on linkages prior to project scale-up has important implications to improve the programme efficiency. All the

implications we have stated are backed by the study findings. The WHO is planning to scale up this programme in Zimbabwe and other Sub-Saharan countries. Similarly, other funding agencies have also shown interest in similar system linkages project. Thus, the programmatic implications listed here are important for programme managers to consider before implementing system linkages project elsewhere.

We prefer to retain the discussion as it was in the previous version. However, we are more than happy to revise if you have any specific comments in the discussion section.

Others:

Comment-11: The authors need to clarify in all stages in the study whether they trying to model factors associated with not linked or not registered for ASRH

Response: Thank you for picking this inconsistency. As mentioned above, we have edited this through out the manuscript with 'not being linked' as the outcome of interest.

Comment-12: Consider deleting lines 231 to 234

Response: We prefer to retain these lines as it details the time of data collection and the process of data collection. As the outcome of our interest is defined with time (linked within 3 months), we feel it is important to highlight the time of data extraction. Also, under the limitation section, we have discussed in detail the bias that would have been introduced due to the way we have extracted data. Thus, we feel it is important to mention about process of data collection under methods section.

Comment-13: Line 270 to 272: The disposition flow chart is unclear without arrows showing directionality. I am not sure if this is because of the BMJ Open system or the way the authors prepared and presented it.

Response: Thank you for highlighting this. We have checked in our submission portal and all the arrows are intact. I think it must be due to BMJ open system.

Comment-14: Line 272 to 273 "the mean (SD) age of study participants was 13.7 (4.3) years": results presented here are not shown in any of the tables Line 285 to 286 "the median (IQR) duration from referral to getting registered at 'youth centre' was 6 (0-56) days": results presented here are not shown in any of the tables.

Response: As highlighted these summary statistics are not shown in any table. Also, we prefer them to be reported only in the text as none of our tables can accommodate these measures.

Comment-15: Line 298 to 299 "Young people referred from the outreach sites and Bulawayo male circumcision clinic had significantly higher rates of not being registered for ASRH services.": Does this refer to the overall or only the sample size considered in the analysis?

Response: Thank you for highlighting this. As mentioned earlier, we have changed this to 'not being linked'. Changes made in line number- 71

Comment-16: Line 315 Reword this sentence to read better “. . . youth centres are however were located . . .” The authors need to be consistent with the referencing style in using () and [].

Response: Thank you for highlighting this typographical error which we had missed. We have edited this now. The changes are made in line number 337 to 339.

Reviewer: 2

Reviewer Name: Makumbi Fred

Institution and Country: Makerere University, Uganda Please state any competing interests or state 'None declared': None declared

Note: All the line numbers mentioned in the response relates to that in the main document with track changes

Comment-1: Abstract lack introduction/background

Response: Thank you for this feedback. The BMJ Open asks for the structured abstract with headings mentioned in the abstract. The BMJ Open doesn't provide option of Introduction/Background for the abstract. As suggested by the reviewer, we have include a line highlighting the recommendation of WHO for establishing linkages between HIV prevention programmes and ASRH services within the objectives section. Also, we have tried to provide brief background about Smart-LyncAges project and rationale for the study. Changes made in line number 48-55.

Comment-2: Did Bulawayo conduct more Outreaches? Please focus on outcome and determine factors associated with that outcome; you now have linked=31.3% but factors are for not linking onto ASRH.

Response: In Bulawayo about 33% of the VMMC clients were circumcised in outreach sites whereas in Mt Darwin it was 31%. The proportion getting circumcised at outreach sites was almost same in both the study districts. The reported relative risks are adjusted for study district and site of circumcision.

The objective of the project was to assess the extent of linkage of young boys from VMMC to ASRH services. Thus, our first objective was to assess the proportion getting 'successfully linked'. For assessing the factors we have considered 'not linked' to ASRH services, as it readily provides the risk groups which the programme has to target for improving the linkage. We have stated this clearly in the objective and wish to retain the same throughout the manuscript

Comment-3: Good to know why the 31% were able to be linked into ASRH

Response: We acknowledge that it would have been ideal if we had qualitatively explored the facilitators and barriers for linkages between the VMMC and ASRH. Due to budgetary and time constraints we failed to explore this. The pathways of linkage were not explored and we have stated this as a limitation of our study (Line number 375-378). However, we have strongly recommended to conduct qualitative study in the main manuscript (Line number 386-392). As the

study findings are disseminated to programme managers, they are keen to conduct the qualitative exploration in near future. This suggestion from the learned reviewer has helped us to push our programme implementers for conducting qualitative study.

Comment-4: Not sure why you should think of 10-14 year old going for ASRH? Could this be reason for low linkage

Response: The ASRH programme in Zimbabwe offers services to young people aged between 10-14 years. The programme assumes that it is critical that early adolescents understand the issues related to HIV and life skills earlier rather than later. Thus, Smart-LyncAges project also had all young people aged 10-24 years as beneficiaries for cross-referral.

Inclusion of 10-14 years could not be the reason for low linkage as about 32% of the participants from 10-14 years age group had been successfully linked through project. In contrary, all the other age groups had relatively lower linkage rate though it was not statistically significant.

Comment-5: What is the picture like for ASRH into VMMC

Response: A study was conducted to assess the linkages between ASRH and VMMC during same reference period. The study showed that about 33% of the those referred from youth centers providing ASRH services reached VMMC clinics. The manuscript describing the linkage between ASRH and VMMC is under review and will be published shortly.

Comment-6: Should have been sent to where the clients wished to get ASRH services

Response: Under the project VMMC clients were referred to their preferred youth centre (mentioned in line number-195-196). Provider would enquire from the adolescent his preferred ASRH centre and referred there. A copy of referral slip was given to peer-educator of the area from which the client has come from. These peer educators were mostly linked to the youth centers preferred by the clients. The referral slips were not sent to youth centres as these centres had no mechanisms to track the referred clients. However, the suggestion made by the reviewer is valid and we would discuss this with programme implementers.

Comment-7: Were the non-eligible referred for the ASRH services? Why or why not?

Response: All the young people registering at VMMC clinic without being circumcised were also referred for ASRH services under the project. The clients are referred to ASRH services during the registration at VMMC clinics. The eligibility for circumcision is assessed later after the registration. Thus, the referral precedes the circumcision eligibility assessment. We have described this in line number 192 to 196. In the study, about 82% of those who were non-eligible for circumcision had been linked to ASRH services.

Comment-8: But what happened?

Response: All the expected staff were available during the study reference period. Thank you for highlighting the lack of clarity in the description given. The statement has been rephrased to portray the correct picture, once again thank you. Changes made in line number 213-214.

Comment-9: What proportion of VMMC were actually from ASRH services?

Response: Of the total, 295 (17%) of the young boys had previously registered for ASRH services (Line number 274 and Figure 1). However, as we did not extract the details of those who had previously registered for ASRH (exclusion criteria), we do not have proportion of clients who were referred through ASRH. Given the fact, the proportion of VMMC clients referred from ASRH services would be less than 17%.

Comment-10: What Percent actually had linkage beyond 3 months? Why not do a time-to-event analysis so you have a chance to include all

Response: None of the participants had been linked beyond 3 months. The maximum duration between referral and registration was 81 days and median (IQR) was 6 (0-56) days.

The time-to-event analysis was not adopted for following reasons, 1) Our interest was to assess the risk groups not getting linked. If time-to-event was adopted we would have ended up modeling for time-to-linkage. Though inverse of HR obtained from the cox-proportional models would have provided the risk groups, it might not have been very intuitive as that of RR. 2) Programatically the implementers were concerned about who are not getting linked rather than who are not getting linked early. As the pathways through peer educators might require relatively greater time to linkage, the associations might have been grossly affected by unmeasured time-dependent confounders. 3) With about 69% not being linked, the right censoring with more than 90 days in 69% would have contributed to erroneous estimates. Even in 69%, the survival time would have varied only because of the date of their registration and inclusion in the cohort. 4) More than 25% of the referred had reached the ASRH services on the same day of referral, this might have over inflated a few of the associations due to unmeasured confounders. Considering these programmatic and mathematical concerns, we decided to adopt event based analysis.

Comment-11: I assume these were 2 independent entrants. But the analysis plan does not indicate if VMMC clients with at least 3 months post registration were follow-up or had some with less than 3 months post VMMC

Response: Yes, these were 2 independent entrants that were merged using the VMMC Number. All the data extraction happened in March, 2019. Data of all clients registered at VMMC clinics during October and November, 2018 was extracted from VMMC 'client intake forms' during March, 2019. Similarly, VMMC-ASRH linkage register for ascertaining the linkage status was assessed in March, 2019. This meant, there was minimum 90 days of follow-up (those registered on November 30th, 2018 was assessed only after March 1st, 2019). We acknowledge that this was not explicitly mentioned in the manuscript and it might create confusion to reader. We sincerely thank the reviewer for highlighting this deficiency. Now, we have added a line to explicitly explaining the minimum follow-up time. Changes made in line number 246-248.

Comment-12: This assume that all considered clients in this analysis were at least 3 months since VMMC.

Response: As explained above, all the participants had a minimum follow-up period of 90 days (~3 months).

Comment- 13: How representative are the sites included in view of all participating sites

Response: The Smart-LyncAges project was implemented only in two districts (Bulawayo and Mt Darwin). We had included all the VMMC clinics of these two districts. Thus, it was a universal sampling with all the project sites restrained by the study reference period.

i.)Very important to know why those who linked were able to link.

Response: As discussed earlier, we did not make an attempt to explore the facilitators and barriers. As secondary data was used in the study, we could not verify qualitatively why clients who were successfully linked were able to get linked. As described earlier, the factors associated with not getting linked was programatically important to identify the risk groups and thus, we modeled with 'not getting linked' as the outcome of interest.

ii) Justify why use log-binomial, and was this model able to converge?

iii) I suggest you use one approach for both bivariate and multivariable regression.

Response: To calculate adjusted RR, two methods are commonly employed – log binomial regression and Poisson regression. Poisson regression model is an acceptable method adapted in statistical software to obtain adjusted RR. Poisson regression with robust variance estimation is applicable in scenarios where there are issues in convergence of log-binomial models used to estimate adjusted RR. Hence, we used Poisson regression. Thank you for highlighting the deficiency in reporting. Now, we have explicitly mentioned the reason for adopting the two different methods under statistical analysis. Also, we are quoting the relevant references (17 and 18) for the same in the revised manuscript. Changes made in line 262 to 270.

iv) It seems your work would have benefited from some interaction, but these were never tested, say between education and age, or rural/urban residence and education check literature review and think through models of health seeking behaviours

Response: As described earlier, the factors associated with not getting linked was programatically important to identify the risk groups and thus, we modeled with 'not getting linked' as the outcome of interest. The models with interaction terms would have been possible if we had done a primary data collection. The secondary data we used was deficient of potential confounders which are essential for building such health seeking behaviour models for linkages. However, as this was not our interest and we limited ourselves to all inclusive explanatory models to find factors associated with 'not being linked'.

Comment-14: More discussion on what worked and why, and what did not why and why

Response: Thank you for this suggestion. As described earlier, we are not well informed about what pathways have worked for establishing linkages. We have explicitly mentioned this under the limitations. We attribute this largely to Smat-LyncAges project as discussed in second paragraph of discussion section. However, this assumption is with the fact that all those linked had produced a reference slip issued from the VMMC clinics. We have also speculated potential reasons for not getting linked under discussion section. To avoid excessive subjective hypothetical speculations in the manuscript, we strongly advocate for qualitative research to find answers for 'Why'.

Comment-15: Yes, rural/urban divide.

Response: Thank you for echoing our thoughts. We now have explicitly mentioned about potential rural-urban divide in linkages. Changes made in line number 305-306.

Comment-16: Need for a service is key. Did you check for if reason for VMMC was associated the seeking ASRH services?

Response: We have checked for potential association between reasons for accessing VMMC (HIV Prevention, Sexual pleasure, STI Prevention, Hygiene and Socio Cultural reasons) and linkage to ASRH services. None of the reasons were significantly associated with seeking ASRH services as shown in Table-3.

Comment-17: This can only be confirmed if you provided precision estimates to see if the 1478 were able to meeting this

Response: We have provided the precision estimate in form of 95% confidence interval for percentage of successfully linked (line number 291-292). The confidence interval (31.3%, 95% CI-29.0%-33.8%) was narrow with the standard error of 1.2%. The relative precision was less than 10% ($2.4/31.3= 8\%$). Thus, as the precision limits can be drawn from the 95% confidence interval, we have made this statement in the manuscript.

Comment-18: Provide a sense of the extent of this problem

Response: Thank you for highlighting this. We have acknowledged the potential underestimation. However, as there are no estimates of the extent of underestimation, we failed to account for it in estimating the percentage of successfully linked. Now, we have stated this explicitly in the manuscript in line number 374-375.

Comment-19: This could be generalized to within those seeking VMMC services. I would assume these to have higher health seeking behaviors anyway but to have only a third link into ASRH, is a low service uptake!

Response: Yes, as mentioned it can be generalised only to VMMC clients within the Smart-LyncAges project districts. Thus, we have retained it as a limitation.

We too acknowledge that the linkage rates could be high as these clients are already in the ambit of health care services. While comparing with other studies we have taken this fact into account. Even the previous studies quoted in manuscript were also assessing the completion of referral from either community health workers or volunteers. However, all the quoted studies had lower linkage rates. Also, we are not trying to make the value judgement only based on the proportion getting linked, but on the possibility of establishing referral systems through these kind of linkage projects. In this context, we feel the linkage rates are promising under Smart-LyncAges project. Though, we state the linkage rates are promising, we strongly recommend for improving the linkages by taking suggested corrective actions.

Comment-20: Was low bearing in mind that these are good health services seeker i.e. the fact they came of VVMC, going to ASRH should have been much higher

Response: As mentioned in the previous response, we agree the linkage rate was low. However, as a model this is promising. Hence, all our implications/recommendations in the paper point towards improving the linkage rates. However, the ASRH services are not perceived as an essential health intervention for the otherwise healthy adolescent boys predominantly in school during most of the year.

Comment-21: Very uncommon for CI to exclude Ho value but p-value to be >5%!

Response: Thank you for highlighting this. As mentioned by the learned reviewer, the CI can never exclude Ho value when the p value is >5%. However, in the given context, the Ho value of '1.0' was seen as the lower limit of confidence interval due to rounding-off of '1.006' to '1.0'. However, now the estimates have changed as minor modification to the model was done.

VERSION 2 – REVIEW

REVIEWER	Kennedy Otwombe Perinatal HIV Research Unit Chris Hani Academic Hospital University of the Witwatersrand Soweto Johannesburg
REVIEW RETURNED	21-Oct-2019

GENERAL COMMENTS	Abstract: The abstract is well summarized. The authors need to clarify what was done and the findings, i.e. one of the objectives was to check for “factors associated with not being linked”, however, in the results part of the abstract “factors associated with not being registered” was reported. Describe the population characteristics in one-line e.g what was their age. Introduction: The authors presented what is happening globally, in Africa then narrowed to Zimbabwe and why the study was conducted. The literature used is relevant and includes recent works. However, the introduction is too long given that only three tables of results have been presented Objectives: The main objectives of the study is stated as assessing the proportion of VMMC clients getting ‘successfully linked’ to ASRH services and factors associated with not being linked. However, the authors keep switching between not linked and not registered for
--

ASRH whereas the data used for final analysis included only those who were not registered for ASRH prior to the study period.

Methodology:

Study design and participants/population

- How was the sample size selected?
- What sampling method was used?
- Eligibility of the participants into study need to be included, for instance, how many were recruited (overall)? How many excluded and why? How many were considered for analysis? Did they all consent? (Line 213 to 216 can be moved under this section)

Measures

- Clearly define all variables/measures described in the table of results
- How were they measured?

Statistical analysis:

- In the statistical methods section, the authors have not described all the statistical methods reported in the tables i.e. mean, median, etc.
- Which variables were considered at the univariate and multivariate levels?
- Clarify why you used RR?
- The multivariate model has a lot of variables. This may lead to collinearity in the model fitted. How was model fit assessed? It appears the authors only reported the full model but a model selection procedure should have been applied to determine the final reduced model.

Results:

- Overall frequencies and percentages presented in table 1 and those not linked in table 3 for all variables considered, I think it will be much easy to read if one table consists of overall, those linked and not linked
- For the proportion of those “successfully linked”: I do not see the need for reporting the confidence interval

Discussion:

The authors present only one page of results and four pages of discussion. It would be helpful to keep the discussion short, clear and linked to the results presented without repetitions.

Overall:

The authors need to clarify in all stages in the study whether they trying to model factors associated with not linked or not registered for ASRH

Consider deleting lines 231 to 234

Line 270 to 272: The disposition flow chart is unclear without arrows showing directionality. I am not sure if this is because of the BMJ Open system or the way the authors prepared and presented it.

Line 272 to 273 “the mean (SD) age of study participants was 13.7 (4.3) years”: results presented here are not shown in any of the tables

Line 285 to 286 “the median (IQR) duration from referral to getting registered at ‘youth centre’ was 6 (0-56) days”: results presented here are not shown in any of the tables

Line 298 to 299 “Young people referred from the outreach sites and Bulawayo male circumcision clinic had significantly higher rates of **not being registered** for ASRH services.”: Does this refer to the overall or only the sample size considered in the analysis?

Line 315 Reword this sentence to read better “. . . youth centres are however were located . . .”

The authors need to be consistent with the referencing style in using () and [].

Additional comments

Major

In the Poisson regression model, it would be helpful for the authors to indicate how they assessed model fit. Whereas they have the benefit of a large sample size, model fit should be assessed and reported to allow the general reader have a better understanding of their model. Additionally, this may identify any collinearity between variables.

	The authors are confusing the interpretation of the model being fitted. They indicate that they are modelling Factors Associated with Not Being Linked. But in reporting their results in the abstract and results section, they report Factors Associated With Linkage to care. Additionally, they are reporting multivariate results but in their results section (in the abstract and results section), they are reporting “independently associated”. These are adjusted results and hence should be reported as adjusted. Independently associated means they are reporting univariate results but the results presented are from the multivariate model. Minor Most of the points I previously raised have been addressed. It would be helpful for the authors to proof read the updated manuscript and address a number of grammatical errors.
--	---

REVIEWER	Fredrick Makumbi School of Public Health Makerere University
REVIEW RETURNED	15-Nov-2019

GENERAL COMMENTS	1) Your outcome was linkage to ASRH services. However, you do not show the key characteristics associated with the primary outcome. I would expect you to do a sub-analysis or expansion of this component, Otherwise one switches a lot between linked and not-linked analysis 2) Your conclusion also has a recommendation "... there is need to explore the reasons for clients not linking..." I know your secondary objective did this. Why would you recommend that? 3) You should present a conclusion on the secondary objective i.e. factors associated with not linkage 4) The key issue to look at given your rich data, would be to conduct a time-to-event analysis. Time-to- successfully linkage into ASRH services
--

VERSION 2 – AUTHOR RESPONSE

Reviewer: 1

Reviewer Name: Kennedy Otwombe

Institution and Country: Perinatal HIV Research Unit, Chris Hani Academic Hospital, University of the Witwatersrand, Johannesburg, South Africa Please state any competing interests or state ‘None declared’: None declared

Major Comments:

Comment-1: In the Poisson regression model, it would be helpful for the authors to indicate how they assessed model fit. Whereas they have the benefit of a large sample size, model fit should be assessed and reported to allow the general reader have a better understanding of their model. Additionally, this may identify any collinearity between variables.

Response: Thank you for highlighting this. We have developed explanatory model with interest of detecting all the possible factors associated with the outcome. As we did not adopt any methods to deduce reduced models, the model improvement and model fit was not commented. However, based on your comment, we calculated the model fit with log likelihood ratio test (LR test) comparing the model with constant. Also, we have mentioned the AIC and BIC values which reflect the model fit. The LR test was significant compared to constant model (p value= 0.014). Changes made in line number-305 to 307.

Comment-2: The authors are confusing the interpretation of the model being fitted. They indicate that they are modelling Factors Associated with Not Being Linked. But in reporting their results in the abstract and results section, they report Factors Associated With Linkage to care. Additionally, they are reporting multivariate results but in their results section (in the abstract and results section), they are reporting “independently associated”. These are adjusted results and hence should be reported as adjusted. Independently associated means they are reporting univariate results but the results presented are from the multivariate model.

Response: Thank you for highlighting this deficiency. We have now consistently used ‘not being linked to ASRH’ as a outcome of interest for our second objective.

We beg to differ here from the suggestion of learned reviewer on ‘independent association’. When a variable is associated with an outcome after adjusting for multiple other potential prognostic factors or confounding variables (often after multivariate-regression analysis), the association is said as an ‘independent association’. It means, the association is independent of effect of confounding variables. So, in our context when we mention ‘independently associated’, it means those which were significant after adjusting for potential confounding variables. I just add a link here for your reference on independently associated- <https://www.evidencepartners.com/glossary/independent-association/>. Thus, we retain the phrase ‘independently associated’ to mean the factors associated with outcome on adjusted analysis.

Minor Comments:

Comment-3: Most of the points I previously raised have been addressed. It would be helpful for the authors to proof read the updated manuscript and address a number of grammatical errors.

Response: Thank you for highlighting the deficiency. We have gone through manuscript and have made required edits.

Reviewer: 2

Reviewer Name: Makumbi Fred

Institution and Country: Makerere University, Uganda Please state any competing interests or state ‘None declared’: None declared

Comment-1: Your outcome was linkage to ASRH services. However, you do not show the key characteristics associated with the primary outcome. I would expect you to do a sub-analysis or expansion of this component, Otherwise one switches a lot between linked and not-linked analysis

Response: Thank you for this suggestion. As stated in the previous reply letter, the intention of the study was to determine the proportion of those referred successfully getting linked to ASRH services and to assess the factors associated with ‘not being linked’. Both the outcomes assessed are programmatically important and has direct implications on the scale-up of the project.

The first objective, provides the extent of linkage between VMMC and ASRH. This provides an insight on the current status of linkage and also helps to set the monitoring targets. The second

objective of assessing the factors associated with 'not getting linked', provides insights on the groups which has high proportion of 'not being linked'. In turn it helps to identify risk groups, which needs to be targeted to improve the linkage. Also, as it is an operational research, our interest here is to simplify the presentation of our findings to easily communicate to policy makers. If we model with 'being linked' as the outcome of interest, we get the groups which has higher proportion of linkage. Thus, we need to follow it up with inverse groups for interpreting the results, which is not intuitive. As we have explicitly mentioned the outcomes in our objectives, we beg to retain the same analysis. However, we are ready to revise the outcomes if the reviewer can specify how the current analysis is harming the interpretation of the findings.

Once we mention the number and proportion of 'not being linked', the inverse of it is 'being linked', as the outcomes are mutually exclusive (Table-3). We felt that, the 'being linked' can be easily deduced from the Table-3. However, based on the learned reviewers comment, we have now added a column of 'linked to ASRH' in Table-3. Changes made in Table-3.

Comment-2: Your conclusion also has a recommendation "... there is need to explore the reasons for clients not linking..." I know your secondary objective did this. Why would you recommend that?

Response: Yes, our conclusion has a strong recommendation for future qualitative study to find out the 'reasons' for 'not getting linked'. The secondary objective just gives insight on 'Who' is not getting linked but doesn't provide idea on 'Why' someone is not getting linked. There is difference between 'Who' and 'Why'. We authors feel, the 'Why' component or the reasons objectively assessed through qualitative study is highly important to institute corrective actions to improve the service. Thus, we add it in the conclusion, to highlight its importance.

Comment-3: You should present a conclusion on the secondary objective i.e. factors associated with not linkage.

Response: Thank you for the suggestion. We have now added a line on the second objective in the conclusion of main manuscript. Changes made in line number 409 and 410.

Comment-4: The key issue to look at given your rich data, would be to conduct a time-to-event analysis. Time-to- successfully linkage into ASRH services

Response: Thank you for the suggestion. However, we the authors feel time-to-event analysis is not programatically and mathematically valid in the current scenario. Just because, we have a data of time-to-event, we don't want to conduct time-to-event analysis. The time-to-event analysis is not adopted for following reasons, 1) Our interest was to assess the risk groups not getting linked. If time-to-event was adopted we would have ended up modeling for time-to-linkage. Though inverse of HR obtained from the cox-proportional models would have provided the risk groups, it might not have been very intuitive as that of RR. 2) Programatically the implementers were concerned about who are not getting linked within three months rather than who are not getting linked early. As the pathways through peer-educators might require relatively greater time to linkage, the associations might have been grossly affected by unmeasured time-dependent confounders. 3) With about 69% not being linked, the right censoring with more than 90 days in 69% would have contributed to erroneous estimates. Even in 69%, the survival time would have varied only because of the date of their registration and inclusion in the cohort. 4) More than 25% of the referred had reached the ASRH services on the same day of referral, this might have over inflated a few of the associations due to

unmeasured confounders. Hope the image below provides better insight to the reviewer on the distribution of the event of interest

With the above distribution, the time-to-event analysis is not valid mathematically. However, we conducted time-to-event analysis and it did not provide any additional insights on the identified risk groups.

VERSION 3 – REVIEW

REVIEWER	Kennedy Otwombe Perinatal HIV Research Unit, Chris Hani Academic Hospital, University of the Witwatersrand, Johannesburg, South Africa
REVIEW RETURNED	09-Dec-2019

GENERAL COMMENTS	One of my core comments on the manuscript was not addressed. Let me provide more details. In regression statistics, one usually fits univariate then multivariate models. A univariate model involves variables that are usually defined as independent or unadjusted. For example, in this manuscript, independent or unadjusted predictors for not getting linked to ASRH services within 3 months amongst others are Bulawayo MC Health Facility (RR: 1.4, 95%CI: 1.3-1.5) and Undergoing Circumcision in Outreach Sites (RR: 1.2, 95%CI: 1.1-1.2). In the adjusted (or multivariate) analysis for these two variables, the RRs are: Bulawayo MC Health Facility (RR: 1.5, 95%CI: 1.3-1.7, $p < 0.0001$) and Undergoing Circumcision in Outreach Sites (RR: 1.2, 95%CI: 1.1-1.3, $p < 0.0001$). The point I want the authors to see is that they cannot then report in their results that independent predictors for Not Getting Linked to ASRH are these two multivariate results. They are not independent but adjusted results.
---

	This is what they have written in the abstract “Receiving a referral from Bulawayo circumcision clinic (RR: 1.5, 95%CI: 1.3-1.7) and undergoing circumcision at outreach sites (RR: 1.2, 95%CI: 1.1-1.3) were independently associated with ‘not being linked’ to ASRH services.” According to results presented in Table 3 under adjusted RR, these findings are not independent but adjusted results. This correction must also be made in the last paragraph of the Results Section i.e line 291. Lines 292-294 on AIC/BIC and LR are not helpful if they don’t tell the readers whether the model is a good fit or not
--	--

REVIEWER	Fredrick Makumbi Makerere University, School of public health Kampala, Uganda
REVIEW RETURNED	02-Dec-2019

GENERAL COMMENTS	i) I have always wondered why the primary outcome is linkage but factors for not linked are assessed? this can be quite confusing ii) This study can benefit from a qualitative study to explore reasons for non-linkage iii) The median age is too low, 13.3years, and so the perception for benefits of ASRH at the point of circumcision may not be fully appreciated thus leading to poor linkage. The study needs to provide more information on the actual messages used at the circumcision sites to ask the clients to link to ASRH services iv) I also note that health facility/health system issues including ASRH provider’s perceptions about ASRH for this young population need to be investigated or highlighted as potential barriers v) If 70% do follow-up post circumcision, indicating good health seeking behaviors, then the 31% reporting for ASRH is too low vi) Lastly I would think that the analysis can benefit from further categorization of the outcome a) Not linked b) Linked within a week (7 days) c) Delayed linkage (8+ days) May be understand these early and delayed linkage may be important
--

VERSION 3 – AUTHOR RESPONSE

Reviewer: 1

Reviewer Name: Kennedy Otvombe

Institution and Country: Perinatal HIV Research Unit, Chris Hani Academic Hospital, University of the Witwatersrand, Johannesburg, South Africa Please state any competing interests or state ‘None declared’: None declared

Comment-1: One of my core comments on the manuscript was not addressed. Let me provide more details. In regression statistics, one usually fits univariate then multivariate models. A univariate model involves variables that are usually defined as independent or unadjusted. For example, in this manuscript, independent or unadjusted predictors for not getting linked to ASRH services within 3 months amongst others are Bulawayo MC Health Facility (RR: 1.4, 95%CI: 1.3-1.5) and Undergoing

Circumcision in Outreach Sites (RR: 1.2, 95%CI: 1.1-1.2). In the adjusted (or multivariate) analysis for these two variables, the RRs are: Bulawayo MC Health Facility (RR: 1.5, 95%CI: 1.3-1.7, $p < 0.0001$) and Undergoing Circumcision in Outreach Sites (RR: 1.2, 95%CI: 1.1-1.3, $p < 0.0001$). The point I want the authors to see is that they cannot then report in their results that independent predictors for Not Getting Linked to ASRH are these two multivariate results. They are not independent but adjusted results. This is what they have written in the abstract "Receiving a referral from Bulawayo circumcision clinic (RR: 1.5, 95%CI: 1.3-1.7) and undergoing circumcision at outreach sites (RR: 1.2, 95%CI: 1.1-1.3) were independently associated with 'not being linked' to ASRH services." According to results presented in Table 3 under adjusted RR, these findings are not independent but adjusted results. This correction must also be made in the last paragraph of the Results Section i.e line 291. Lines 292-294 on AIC/BIC and LR are not helpful if they don't tell the readers whether the model is a good fit or not

Response: Thank you for this detailed comment. However, we request the reviewer to go through this web link to understand the meaning of word 'independent association-
<https://www.evidencepartners.com/glossary/independent-association/>. The word 'independent association' is used to denote the associations after adjusting for potential confounders. Which means, when a variable is associated with an outcome after adjusting for other potential prognostic factors or confounding variables (through multivariate-regression analysis), the association is said as an 'independent association'. Please note, there is a difference between 'independent variable' and 'independently associated'. The 'independent variables' are all those which are used to explain the dependent or outcome variable in either univariate or multivariate models. The 'independently associated' means the variable is associated with the outcome after adjusting for confounding variables. So, when we mention 'independently associated' it doesn't mean the association is from univariate analysis as understood by the learned reviewer.

Though we don't agree with the suggestion, to avoid further delay in processing of the manuscript, we have made changes as suggested by the reviewer in the abstract and results section. Now we have reported as 'On adjusted analysis, receiving referral from Bulawayo circumcision clinic (aRR-1.5 (95% CI-1.3-1.7)) and undergoing circumcision at outreach sites (aRR-1.2 (95% CI-1.1-1.3)) were associated with 'not being linked' to ASRH services.' Changes made in line number 66 to 69 and 291 to 293.

As the learned reviewer might be aware, the 'goodness of fit' can not be commented when the generalised linear models with modified Poisson regression is used. The best possible metrics for commenting on the quality of the model is to describe the AIC, BIC and LR test comparing with the constant. Thus, we have adopted the later to describe the quality of the model. We would be happy to deduce the 'goodness-of-fit' if the reviewer can suggest on the test or method to do so.

Reviewer: 2

Reviewer Name: Makumbi Fred

Institution and Country: Makerere University, Uganda. Please state any competing interests or state 'None declared': None declared

Comment-1: I have always wondered why the primary outcome is linkage but factors for not linked are assessed? this can be quite confusing

Response: Thank you for this suggestion. As stated in the previous reply letter, the intention of the study was to determine the proportion of those referred successfully getting linked to ASRH services and to assess the factors associated with 'not being linked'. Both the outcomes assessed are programmatically important and has direct implications on the scale-up of the project.

The first objective, provides the extent of linkage between VMMC and ASRH. This provides an insight on the current status of linkage and also helps to set the monitoring targets. The second objective of assessing the factors associated with 'not getting linked', provides insights on the groups which has

high proportion of 'not being linked'. In turn it helps to identify risk groups, which needs to be targeted to improve the linkage. Also, as it is an operational research, our interest here is to simplify the presentation of our findings to easily communicate to policy makers. If we model with 'being linked' as the outcome of interest, we get the groups which has higher proportion of linkage. Thus, we need to follow it up with inverse groups for interpreting the results, which is not intuitive. As we have explicitly mentioned the outcomes in our objectives, we beg to retain the same analysis as the outcomes chosen are programmatically relevant.

Comment-2: This study can benefit from a qualitative study to explore reasons for non-linkage

Response: Thank you for this suggestion. We completely agree with the reviewers comment that it would have been ideal if a qualitative component was added. We have explicitly mentioned the need for qualitative study in our recommendations and conclusion. Now, we are happy to inform the reviewer that the qualitative study exploring the deficiencies in linkages is being conducted in the study setting. The ethics approvals have been obtained and interviews are being conducted among VMMC service providers, ASRH service providers, parents and adolescents. We thank the learned reviewer as the similar comment for qualitative study made during the first round of revision was used to convince programme leads to allocate budget for conduct of qualitative study. We will be able to draft the manuscript with qualitative data by June 2020.

Comment-3: The median age is too low, 13.3 years, and so the perception for benefits of ASRH at the point of circumcision may not be fully appreciated thus leading to poor linkage. The study needs to provide more information on the actual messages used at the circumcision sites to ask the clients to link to ASRH services.

Response: Thank you for this suggestion. The ASRH programme in Zimbabwe offers services to young people aged between 10-24 years. The programme assumes that it is critical that early adolescents understand the issues related to HIV and life skills earlier rather than later. Thus, Smart-LyncAges programme also had all young people aged 10-24 years as beneficiaries for cross-referral. Also, as most of adolescent seek VMMC services at early adolescent age, the median age is 13.3 years. We don't feel that low age is a reason for poor linkage as the age group 10-14 years (32%) had better successful linkage rate compared to 20-24 years (26%). However, we agree to the suggestion on highlighting the details of counselling. We have added details of counselling on ASRH services at VMMC clinic. Changes made in line number 189 to 191.

Comment-4: I also note that health facility/health system issues including ASRH provider's perceptions about ASRH for this young population need to be investigated or highlighted as potential barriers.

Response: Thank you for the suggestion. As mentioned earlier, we are conducting qualitative study for exploring the barriers for successful linkages to ASRH services. As suggested, we are interviewing the healthcare providers at VMMC clinics and ARSH services. Thus, we hope we will be able to get insights on ASRH provider's perceptions during qualitative study. However, we feel this comment is beyond the scope of this manuscript.

Comment-5: If 70% do follow-up post circumcision, indicating good health seeking behaviors, then the 31% reporting for ASRH is too low.

Response: Thank you for this comment. We agree with the reviewer that the study group showed a good health seeking behaviour in terms of attending post-circumcision follow-up care. However, we beg to differ with the opinion of the reviewer on judging the rate of linkage in comparison with the percentage seeking post-circumcision follow-up care. The adolescent value post circumcision follow-up high as they need wound care, which is an immediate necessity due to pain associated with raw wound and healing. Whereas, as discussed in the manuscript, the preventive strategy like ASRH may not be considered as an immediate need by young people. Thus, we can't make a value judgement on the rate of linkages in comparison with post-circumcision follow-up rate. However, the high rate of post-circumcision follow-up is encouraging as this can be used to reinforce the need for ASRH services to improve linkages. In line number 385 to 388, we have stated this as an implication and have recommended utilizing this opportunity to improve linkages.

Comment-6: Lastly, I would think that the analysis can benefit from further categorization of the outcome

- a) Not linked
- b) Linked within a week (7 days)
- c) Delayed linkage (8+ days)

Response: Thank you for this suggestion. However, as stated in the manuscript, the interest is to assess the factors associated with 'not being linked within three months'. The linkage within three months is the defined programmatic outcome which the implementers use for monitoring the programme. Thus, we need to assess the risk groups for not getting linked within three months for informing the programme managers. As the ASRH services are not immediate necessity, the programme is not interested to see how early someone is getting linked. Breaking the outcome into three groups brings several challenges in adjusted ordinal regression analysis and interpretation of the results. Also, the sample size and power for carrying out such analysis is not adequate for carrying out such analysis. Thus, as the outcomes and the programmatic importance of the selected outcomes are clearly defined in the manuscript, we wish to adhere to this analysis.